# DESIGNING CONCISE CONVNETS WITH COLUMNAR STAGES

**Ashish Kumar**
ScoreLabsAI
Atlanta, USA
ashishkumar@gmail.com

**Jaesik Park**[*]
Seoul National University
Seoul, South Korea
jaesik.park@snu.ac.kr

## ABSTRACT

In the era of vision Transformers, the recent success of VanillaNet shows the huge potential of simple and concise convolutional neural networks (ConvNets). Where such models mainly focus on runtime, it is also crucial to simultaneously focus on other aspects, e.g., FLOPs, parameters, etc, to strengthen their utility further. To this end, we introduce a refreshing ConvNet macro design called **Co**lumnar **St**age **Net**work (CoSNet). CoSNet has a systematically developed simple and concise structure, smaller depth, low parameter count, low FLOPs, and attention-less operations, well suited for resource-constrained deployment. The key novelty of CoSNet is deploying parallel convolutions with fewer kernels fed by input replication, using columnar stacking of these convolutions, and minimizing the use of $1 \times 1$ convolution layers. Our comprehensive evaluations show that CoSNet rivals many renowned ConvNets and Transformer designs under resource-constrained scenarios. Code: https://github.com/ashishkumar822/CoSNet.

## 1 INTRODUCTION

In the past decade, there has been enormous study in the neural network architectures Krizhevsky et al. (2012); Simonyan & Zisserman (2014), demonstrating that different information paths He et al. (2016); Huang et al. (2017); Szegedy et al. (2015); Tan & Le (2019); Xie et al. (2017) can affect the performance. However, as highlighted in recent VanillaNet Chen et al. (2023), due to the increased network complexity, the primary source of runtime bottleneck would be the off-chip memory traffic apart from the main computations because GPUs are constantly becoming more powerful.

The issue is prevalent in more advanced models, such as ConvNext Liu et al. (2022), CoatNet Dai et al. (2021b), ViT Dosovitskiy et al. (2020), etc., due to the indirect information paths or the attention mechanism that requires frequent memory reordering. Hence, despite these models being far ahead of their simpler counterparts He et al. (2016); Krizhevsky et al. (2012), there are still opportunities to develop concise models for better accuracy, runtime, and resource tradeoffs.

Efforts in this direction are noteworthy. For example, RepVGG Ding et al. (2021) improves runtime via structural parameterization. ParNet Goyal et al. (2021) reduces depth by utilizing multiple shallower network modules. Recent VanillaNet Chen et al. (2023) merges layers during inference while avoiding branches. These works fall in the paradigm of simplifying ConvNet models for resource-constrained scenarios, in contrast to the advanced ConvNets Dai et al. (2021b); Liu et al. (2022), or ViT Dosovitskiy et al. (2020) focusing on state-of-the-art accuracy.

We are inspired by the utility of the former class of works, i.e., simpler and concise models. However, besides focusing on runtime or depth Chen et al. (2023); Ding et al. (2021); Goyal et al. (2021), we also focus on other ConvNet aspects, such as FLOPs, parameters, depth, computational density, etc. To this end, we propose a concise model by revisiting the fundamentals of prominent ConvNet designs and define the following key sub-objectives:

1) *Reducing depth:* Network depth refers to the number of layers stacked. More depth means more sequential operations, thus more latency and wastage of parallel computing elements (GPU cores).
2) *Controlled parameter growth:* Reducing depth to achieve lower latency leads to an increased number of parameters Chen et al. (2023); Goyal et al. (2021), thus necessitating parameter control

---

[*]Corresponding author

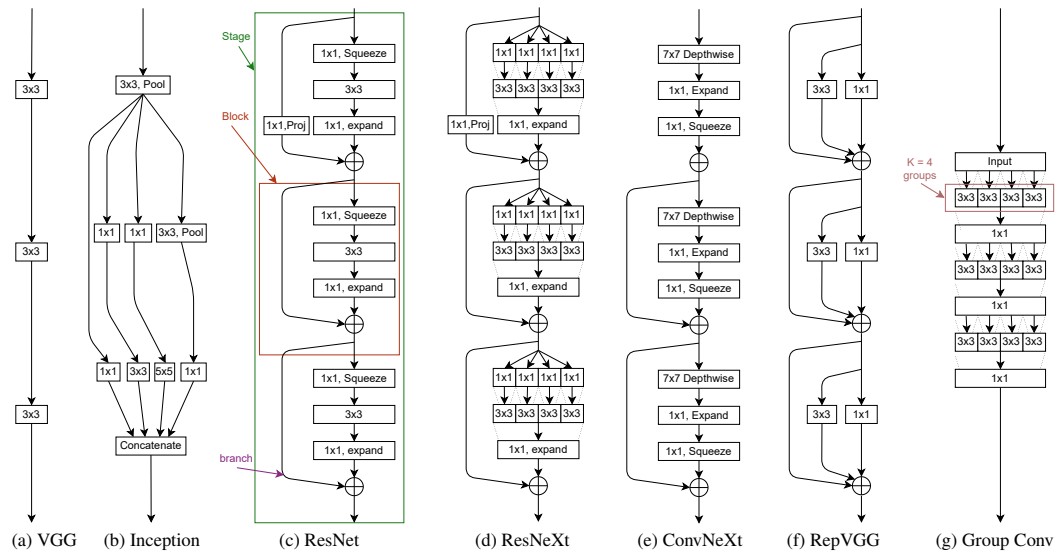

Figure 1: Design of various representative architectures in the order of their development in the timeline from (a) to (e). Each graph represents a stage of a network operating at a particular resolution.

while having short depth.

3) *Low branching:* Network branching increases memory requirements to hold intermediate tensors and also increases memory access cost to account for the branched operations.

4) *High computational density:* A layer must have a high computing density since fewer computations per layer waste the parallel computing cores, e.g., depthwise convolutions Howard et al. (2017) have less computation density and high memory access cost compared to the dense convolutions Simonyan & Zisserman (2014).

5) *Uniform primitive operations:* Maintaining a uniform convolution kernel size throughout the network and branches is desirable so that computations can be packed into minimum GPU transactions.

This leads to a concise refreshing ConvNet design (Figure 2) that shows enhanced performance in various aspects, such as low memory consumption, low memory access costs on parallel computing hardware, smaller depth, minimum branching, lower latency, low parameter count, and reduced FLOPs. The key attributes of CoSNet-unit are *parallel columnar convolutions* (Sec. 3.2), *input replication* (Sec. 3.3), and *shallow-deep projections* (Sec. 3.7), allowing CoSNet to perform better than simple ConvNets or rival the advanced designs. The achievements of CoSNet emphasize simplicity's importance in effective ConvNet designs.

## 2 RELATED WORK

This section provides an overview of representative network designs (Figure 1). The earlier ConvNets (Krizhevsky et al., 2012; Simonyan & Zisserman, 2014) stacked dense convolutions with an increasing number of channels and decreasing resolution (Figure 1a). Improved versions (He et al., 2016; Szegedy et al., 2015; Xie et al., 2017) achieve higher accuracy via manually designed blocks (Figure 1c), while (Howard et al., 2017; Ma et al., 2018; Sandler et al., 2018; Zhang et al., 2018), use depthwise convolutions (Sifre & Mallat) for saving computations, but they are not memory friendly (Ding et al., 2021).

ConvNets have also grown from branchless (Krizhevsky et al., 2012; Simonyan & Zisserman, 2014) (Figure 1a) to single branch (He et al., 2016) (Figure 1c) to multi-branch (Radosavovic et al., 2020; Szegedy et al., 2016; Tan & Le, 2019; Zoph et al., 2018) (Figure 1b). These models utilize $1 \times 1$ convolutions frequently, which rapidly increases network depth (He et al., 2016; Sandler et al., 2018; Tan & Le, 2019; Zhang et al., 2018) (Figure 1c-1e). Although beneficial, both large depth and high branching tend to increase the latency, memory requirements, and Memory Access Cost (MAC) (Chen et al., 2023) due to the serialized execution of parallel branches (Ding et al., 2021; Srivastava et al., 2015; Tan & Le, 2019).

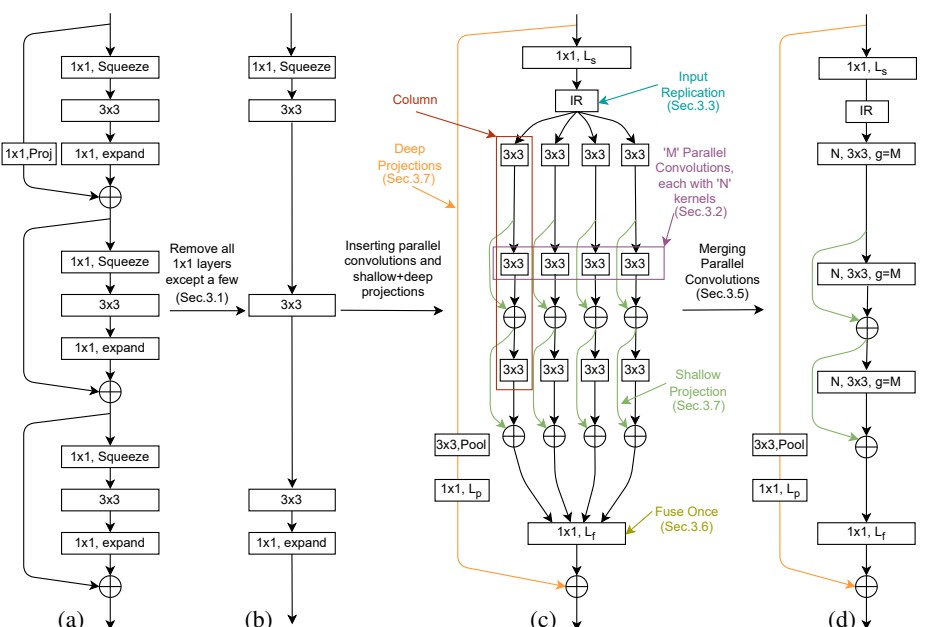

Figure 2: Design evolution flow of CoSNet-unit. (a) A ResNet (He et al., 2016) stage with three blocks. (b) removing all $1 \times 1$ convolutions except the first of the first block and the last of the last block. (c) detailed design of the CoSNet-unit by integrating our design ideas into '(b)', and (d) final optimized CoSNet-unit from an implementation viewpoint.

Recent RepVGG (Ding et al., 2021) proposes structural parameterization (SR) to resolve the branching issue. While ParNet (Goyal et al., 2021) and VanillaNet (Chen et al., 2023) reduce depth to achieve lower latency. Efforts to reduce depth increase the parameter count (Chen et al., 2023; Goyal et al., 2021) to match the accuracy of relatively deeper counterparts (He et al., 2016).

Recent Vision Transformers (ViTs) (Dosovitskiy et al., 2020; Liu et al., 2023; 2021; Touvron et al., 2021) have attracted huge research interests. As outlined in (Dai et al., 2021b), the $O(N^2)$-complex attention in ViTs is a notable issue from a data size and resource-constrained viewpoint. This issue continues to inspire improvements in ConvNets. For instance, RepLKNet (Ding et al., 2022) aims to bridge the gap between ViT and CNNs by employing large kernels.

The above designs focus on limited aspects, e.g., (He et al., 2016; Xie et al., 2017) on the accuracy, (Chen et al., 2023; Goyal et al., 2021) on runtime and depth. To address this research gap, we draw inspiration from the success of VanillaNet-style networks, and instead of pursuing large-scale models, we focus on our sub-objectives (Sec 1) and revisit the representative ConvNets to push the frontier of simple, concise models.

## 3 COLUMNAR STAGE NETWORK

Our approach is a series of improvements motivated by representative ConvNet designs. To understand better, we begin with ResNet (He et al., 2016) as a stepping stone as done in (Liu et al., 2022). We design the building block of CoSNet i.e., CoSNet-unit while recalling our sub-objectives: 1) reducing depth, 2) controlled parameter count, 3) high computational density, 4) uniform primitive operation, and 5) low branching.

### 3.1 AVOIDING $1 \times 1$ FOR REDUCING DEPTH

The recent works of reducing depth (Chen et al., 2023; Goyal et al., 2021) increase the parameter count to achieve accuracy similar to a deeper network. However, we aim to reduce depth while avoiding a large parameter count, which is a difficult objective. Hence, we handle reducing depth and controlling parameter count separately.

To reduce depth, we identify that $1 \times 1$ convolutions in the ResNet-like designs (Figure 2a) (He et al., 2016; Liu et al., 2022) etc., form almost 66% of depth without improving receptive field due to their pointwise nature (Luo et al., 2016). Hence, we minimize the number of these layers. Specifically, we use only two $1 \times 1$ convolutions $L_s$ and $L_f$ in a CoSNet-unit, where $L_s$ reduces the channel squeezing while $L_f$ performs expansion (Figure 2b). Then, we stack $l$ number of $3 \times 3$ convolutions, forming a *column* sandwiched between $L_s$ and $L_f$.

This strategy brings two benefits. *First*, it reduces the overall depth at the same receptive field, e.g., three blocks of ResNet-like design have 9 layers with three receptive-field governing $3 \times 3$ layers. In contrast, the proposed design only has 5 layers, i.e., two $1 \times 1$ and three $3 \times 3$ conv, indicating a notable 45% *depth reduction* with the same receptive field.

*Second*, the reduced depth results in *reduced FLOPs* and *latency* e.g., CoSNet performs better than ResNet-50 at 50% fewer layers while having relatively fewer parameters, FLOPs, and latency.

## 3.2 PARALLEL COLUMNAR CONVOLUTIONS FOR CONTROLLED PARAMETERS.

We propose *Parallel Columnar Convolutions* to handle the large parameter count originating to compensate for the lost non-linearity due to the reduced depth (Chen et al., 2023). In this design, we first deploy $M$ columns in parallel (Figure 2c), and crosstalk among columns does not exist, i.e. a convolution of a column can only feed a convolution of the same column. Then, we restrict the number of kernels in a convolution layer of a column to a small number of $N$. This design affects the number of parameters less aggressively when the number of columns increases (see ablations in the supplement). This is a powerful feature of CoSNet design, offering controlled growth of parameter count during network scaling. This helps CoSNet achieve higher accuracy with fewer parameters.

The idea of the parallel column is based on our hypothesis that multiple kernels with fewer channels can be better than one with large channels. Having $M$ convolutions in parallel with a smaller number of kernels $N$ is equivalent to synthesizing multiple kernels from a large kernel. On the other hand, the idea of smaller $N$ is motivated by the fact that many parallelly operating neurons tend to learn redundant representations while being computationally taxing and causing overfitting. For the same reason, EfficientViT (Liu et al., 2023) slices the input channels in its structure. Hence, by keeping $N$ small, we expect to decouple the data patterns learned by the different columns.

In ConvNets, a similar idea was proposed in Inception (Szegedy et al., 2015), then in ResNeXt (Xie et al., 2017), and then abandoned later as it caused inefficiency. For instance, Inception uses different-sized convolutions and pooling in parallel, which must be executed serially despite being employed in parallel. Also, Inception differs from our columnar architecture since it does not have columns as deep as CoSNet.

## 3.3 INPUT REPLICATION

In CoSNet, all the columns are fed with replicas of the input. We achieve that via a simple *Input Replication* IR operation (Figure 2c), which transforms a tensor $\in \mathbb{R}^{C \times H \times W}$ into duplicated one $\in \mathbb{R}^{(M \times C) \times H \times W}$, where $M$ denotes the desired number of the columns. In the CoSNet-unit, the IR is applied over the output of the $L_s$ layer to feed each column with the input replica.

Input replication has also been employed in the earlier ResNeXt (Xie et al., 2017), but notable differences exist. ResNeXt has multiple blocks per stage, and *each block performs* IR, as shown in Figure 1d. Whereas CoSNet performs IR only once. In ResNeXt, IR is performed before $1 \times 1$ squeeze layer, whereas in CoSNet, it is done after the squeeze layer.

The parallel columnar organization may seem to overlap with widely explored group convolutions (Xie et al., 2017; Zhang et al., 2018). However, there are two key differences. *First*, group convolution *divides the input channels*, thus defying the objective of IR because now each column receives only a subset of the input channels, thus less information per group, as shown in Figure 1g. On the contrary, CoSNet uses IR, which feeds each column with the replica of the input, thus making the entire input information accessible to each column. This becomes one of the reasons that despite infrequent fusion (Sec. 3.6), unlike group conv, CoSNet still performs better (See ablations in the supplement).

### 3.4 UNIFORM KERNEL SIZE FOR HIGH COMPUTATIONAL DENSITY & UNIFORM PRIMITIVE OPERATIONS.

The parallel columns of a CoSNet-unit can be executed independently; however, this design can be optimized further if all the convolutions in all the columns have uniform kernel size. To this end, we first set the kernel size in all the convolutions to $k \times k$, where $k \in \mathbb{R}_{\geq 3}$. Then, we combine the convolutions of different columns lying at the same level, i.e., the first convolution of each column is combined into one convolution having $M$ batches.

With this optimization, all columns (Figure 2c) can be efficiently processed using GPU-based highly optimized Batched-Matrix-Multiply routines, leading to increased computational density, increased GPU utilization, reduced memory access cost (Ding et al., 2021), and minimized GPU load-dispatch transactions. Thus resulting in a simplified CoSNet design (Figure 2d). Moreover, since an CoSNet-unit is made up mostly of $3 \times 3$ convolutions, it well suits the convolution hardware accelerators because they have dedicated support for them, and more chip area can be dedicated to $3 \times 3$ computational units.

### 3.5 BATCHED PROCESSING FOR MINIMAL BRANCHING.

From the previous step, batched processing yields additional benefits, i.e., CoSNet becomes uni-branched regardless of training and testing. This reduces memory consumption and access costs, resulting in lower per-iteration training time and increased parallelization. This contrasts with RepVGG (Ding et al., 2021), which has a considerable training time. Regarding ASIC development, low branching in CoSNet leaves more area on the chip because of the reduced memory requirement to store intermediate tensors. This area can now be dedicated to more computational units.

Although the multi-branch design is beneficial for achieving high accuracy (Ding et al., 2021) (Figure 1f), CoSNet, despite having minimal branching, effortlessly achieves high accuracy. This is because the core design of CoSNet-unit posses multiple branches in the form of columns and short projections (Figure 2c). However, due to batched processing CoSNet-unit mimics uni-branched behavior. In this way, CoSNet takes advantage of both worlds, i.e., eliminated train time complexity due to multiple branches and fast inference during test time without needing structural parameterization (Ding et al., 2021).

### 3.6 FUSE ONCE

Finally, the output of all the columns is fused by $L_f$. In ResNeXt (Figure 1d), the output of $3 \times 3$ convs are fused immediately via a $1 \times 1$ conv, whereas in CoSNet, it is done much later. Our fuse once strategy is different from group (Zhang et al., 2018) or depthwise convolutions (Howard et al., 2017) that are followed by $1 \times 1$ (Figure 1g) to avoid loss of accuracy because each group/channel has too few connections which restrict its learning ability without frequent fusion ((Zhang et al., 2018), Figure 1g). This increases network depth and, hence, latency. On the contrary, CoSNet is free from this constraint because we increase $N$ as we go deep in CoSNet unlike (Zhang et al., 2018). Hence, each neuron in $M$ columns has a sufficiently large number of connections that enable learning without frequent fusion. We performed an ablation (see supplement) by applying the same strategy as Figure 1g in CoSNet. We observed increased network depth, latency, and decreased accuracy.

*Pairwise Frequent Fusion (PFF):* Although we aim to reduce $1 \times 1$ layers as they have a high concentration of most of the network parameters and FLOPs (Sec. 3.1), we propose a frequent fusion scheme via $1 \times 1$ while avoiding the parameter and FLOPs concentration issue. In this scheme, instead of fusing all the columns simultaneously, we fuse columns only pairwise via $1 \times 1$ (Figure 3). This strategy essentially offers several benefits. Firstly, with pairwise fusion, $1 \times 1$ kernel incorporates only a few computations per layer due to small kernel size (fewer channels) while improving network accuracy. Secondly, the latency incorporated due to these layers does not increase the overall latency because of the few computations per

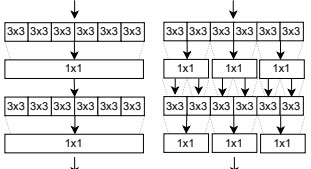

Figure 3: Illustration of Vanilla Frequent Fusion (left) ((Zhang et al., 2018), Figure 1g) and Pairwise Frequent Fusion (right).

layer, hence offers better accuracy with negligible latency overhead ($1 - 2$ms). See Table 1. We denote all such CoSNet variants as CoSNet-PFF.

### 3.7 PROJECTIONS

To facilitate better gradient flow during network training, we employ projections introduced by ResNet (He et al., 2016) but slightly differently in two ways:

1) *Shallow Range.* These projections are formed between any two layers of a column and promote better gradient flow through the stack of $l$ layers (Figure 2c). Since such projections connect only two layers, unlike a stack of layers in ResNet-like designs, these are named shallow ranges.

2) *Deep Range.* These projections are formed between the input and the output of a CoSNet-unit. Specifically, the input to CoSNet-unit is projected to its output via a $3 \times 3$ pooling layer followed by a $1 \times 1$ convolution $L_p$ whose output is fused with the output of $L_f$ (Figure 2c). The pooling operation gathers spatial context by enlarging the receptive field, which is otherwise impossible for $L_p$ alone due to its point-wise nature. We call it deep projection because it bypasses the entire columnar structure while combining information from the previous network stages, i.e., multi-layer information fusion, and providing a short alternative path for gradient flow.

The above projection design helps achieve CoSNet better accuracy (see ablations) and is slightly different from the existing ones. First, projection in ResNet-like models (He et al., 2016; Xie et al., 2017) is used only in the first block of a stage (shallower), and projection between stages does not exist. Second, projection in these models operates at a stride of 2. On the contrary, in CoSNet, the projection connects two stages (deeper) while operating at unit stride and utilizing pooling to increase the receptive field.

### 3.8 COSNET INSTANTIATION

A CoSNet variant can be instantiated by stacking CoSNet-units (Figure 4). CoSNet does not have the notion of blocks but only has stages in the form of CoSNet-unit. This contrasts with existing ConvNets, which have stages, and each stage comprises multiple blocks (Goyal et al., 2021; He et al., 2016; Liu et al., 2022; Xie et al., 2017) e.g., ResNet-50 has four stages, having 3, 4, 6, and 3 blocks respectively (Figure 1c- 1e).

To instantiate a CoSNet variant, we follow the tradition of five stages (He et al., 2016; Simonyan & Zisserman, 2014), among which the first (stem) is a $3 \times 3$ convolution with a stride of 2, while the remaining are the CoSNet-units.

Following ResNet (He et al., 2016), we set channels of $L_s$ to 64, which gets doubled at each stage, while the channels of $L_p$ and $L_f$ always equal to $\zeta$ times the channels of $L_s$. We set $\zeta = 4$, following (He et al., 2016). To further simplify the instantiation,

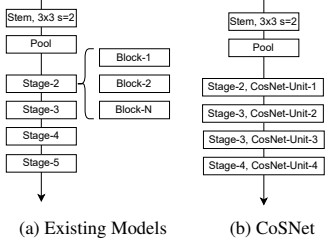

(a) Existing Models    (b) CoSNet

Figure 4: Macro design of (a) existing networks e.g. Ding et al. (2021); He et al. (2016); Liu et al. (2022); Xie et al. (2017), and (b) CoSNet. CoSNet does not have blocks in its stages.

we set the depth of a column, i.e., $l$ in $k^{th}$ CoSNet-unit equal to the number of blocks in the $k^{th}$ stage of a widely used model ResNet-50 (He et al., 2016). Summarily, CoSNet-unit has *only three hyperparameters: $M$, $N$, $l$* which control CoSNet's parameters, depth, latency, and accuracy. Hence, different CoSNet variants can be constructed by changing them. Please refer to the supplement for CoSNet instance names and ablations on $M$, $N$, $l$.

## 4 EXPERIMENTS

We evaluate CoSNet on ImageNet (Deng et al., 2009) dataset consisting of 1.28M train and 50k validation images of 1000 categories. Our training methodology is consistent with recent VanillaNet (Chen et al., 2023). We use data augmentation techniques in (Chen et al., 2023; Liu et al., 2022). See the appendix at the end of this paper for more details.

### 4.1 ADVANCED CONVNETS AND VISION TRANSFORMERS

**CoSNet *vs* recent EfficientViT (Liu et al., 2023)** As shown in Table 1 and Figure 5, CoSNet is less deep and runs 60% faster than EfficientViT Transformer while exhibiting better accuracy, e.g., EfficientVit-M4 vs CoSNet-A0. EfficientVit is another example of lower FLOPs that do not guarantee

Table 1: Evaluation of CoSNet on ImageNet Deng et al. (2009). Latency is measured with batch size 1. 'SR' denotes structural parameterization. 'PFF' stands for pairwise frequent fusion. See Sec 3.6 for details.

| Architecture | Type | #Depth ↓ | #Params ↓ | FLOPs ↓ | Latency ↓ | FPS ↑ | Top-1 (%) ↑ |
|---|---|---|---|---|---|---|---|
| ResNet-18 He et al. (2016) | ConvNet | 18 | 11.6M | 1.83B | 4ms | 250 | 71.1 |
| ResNet-34 He et al. (2016) | ConvNet | 34 | 21.7M | 3.68B | 8ms | 125 | 74.1 |
| ResNet-50 He et al. (2016) | ConvNet | 50 | 25.5M | 4.12B | 11ms | 90 | 76.3 |
| ResNet-101 He et al. (2016) | ConvNet | 101 | 44.5M | 7.85B | 15ms | 67 | 77.2 |
| ResNet-152 He et al. (2016) | ConvNet | 152 | 60.1M | 11.50B | 15ms | 67 | 77.8 |
| ResNeXt-50 Xie et al. (2017) | ConvNet | 50 | 25.1M | 4.40B | 11ms | 90 | 77.4 |
| ResNeXt-101 Xie et al. (2017) | ConvNet | 101 | 44.1M | 8.10B | 14ms | 71 | 78.4 |
| EfficientNet-B0 Tan & Le (2019) | ConvNet | 49 | 5.3M | 0.40B | 8ms | 125 | 75.1 |
| RegNetX-12GF Radosavovic et al. (2020) | ConvNet | 57 | 46.0M | 12.10B | 13ms | 77 | 80.5 |
| RepVGG-A0 Ding et al. (2021) | ConvNet | 22 | 8.3M | 1.46B | 4ms | 250 | 72.4 |
| RepVGG-A0 Ding et al. (2021) w/o SR | ConvNet | 22 | 9.1M | 1.51B | 8ms | 125 | 72.4 |
| RepVGG-A1 Ding et al. (2021) | ConvNet | 22 | 12.7M | 2.36B | 5ms | 200 | 74.4 |
| RepVGG-A1 Ding et al. (2021) w/o SR | ConvNet | 22 | 14.0M | 2.63B | 7ms | 143 | 74.4 |
| RepVGG-B0 Ding et al. (2021) | ConvNet | 28 | 14.3M | 3.40B | 5ms | 200 | 75.1 |
| RepVGG-B0 Ding et al. (2021) w/o SR | ConvNet | 28 | 15.8M | 3.06B | 7ms | 143 | 75.1 |
| RepVGG-A2 Ding et al. (2021) | ConvNet | 22 | 25.5M | 5.12B | 7ms | 143 | 76.4 |
| RepVGG-A2 Ding et al. (2021) w/o SR | ConvNet | 22 | 28.1M | 5.69B | 9ms | 111 | 76.4 |
| RepVGG-B3 Ding et al. (2021) | ConvNet | 28 | 110.9M | 26.20B | 17ms | 58 | 80.5 |
| RepVGG-B3 Ding et al. (2021) w/o SR | ConvNet | 28 | 123.0M | 29.10B | 22ms | 45 | 80.5 |
| ParNet-L Goyal et al. (2021) | ConvNet | 12 | 55.0M | 26.70B | 23ms | 43 | 77.7 |
| ParNet-XL Goyal et al. (2021) | ConvNet | 12 | 85.0M | 41.50B | 25ms | 40 | 78.5 |
| DeiT-S Touvron et al. (2021) | Transformer | 48 | 22.0M | 4.60B | 15ms | 66 | 79.8 |
| Swin-T Liu et al. (2021) | Transformer | 96 | 28.0M | 4.50B | 20ms | 50 | 81.1 |
| ViTAE-S Xu et al. (2021) | Transformer | 116 | 23.6M | 5.60B | 24ms | 41 | 82.0 |
| CoAtNet-0 Dai et al. (2021b) | Hybrid | 64 | 25.0M | 4.20B | 15ms | 66 | 81.6 |
| ConvNeXt-T Liu et al. (2022) | ConvNet | 59 | 29.0M | 4.50B | 13ms | 77 | 81.8 |
| ConvNextV2-P Woo et al. (2023) | ConvNet | 41 | 9.1M | 1.37B | 11ms | 90 | 79.7 |
| ConvNextV2-N Woo et al. (2023) | ConvNet | 47 | 15.6M | 2.45B | 13ms | 77 | 81.2 |
| ConvNextV2-T Woo et al. (2023) | ConvNet | 59 | 28.6M | 4.47B | 16ms | 62 | 82.5 |
| EfficientViT-M4 Liu et al. (2023) | Transformer | 42 | 8.8M | 0.30B | 6ms | 166 | 74.3 |
| EfficientViT-M5 Liu et al. (2023) | Transformer | 70 | 12.4M | 0.60B | 7ms | 142 | 76.8 |
| VanillaNet-6 Chen et al. (2023) | ConvNet | 6 | 32.0M | 6.00B | 6ms | 167 | 76.3 |
| VanillaNet-8 Chen et al. (2023) | ConvNet | 8 | 37.1M | 7.70B | 6ms | 167 | 79.1 |
| VanillaNet-9 Chen et al. (2023) | ConvNet | 9 | 41.4M | 8.60B | 6ms | 167 | 79.8 |
| VanillaNet-10 Chen et al. (2023) | ConvNet | 10 | 45.7M | 9.40B | 7ms | 142 | 80.5 |
| InceptionNeXt-S (Yu et al., 2024) | ConvNet | 48 | 49.0M | 8.40B | 18ms | 55 | 83.5 |
| UniRepLKNet-S (Ding et al., 2024) | ConvNet | 180 | 56.0M | 9.10B | 23ms | 43 | 83.9 |
| • **CoSNet-A**0 | ConvNet | 26 | 8.8M | 1.25B | 6ms | 167 | 77.1 |
| • **CoSNet-A**1 | ConvNet | 26 | 12.1M | 1.70B | 6ms | 167 | 78.2 |
| • **CoSNet-B**0 | ConvNet | 26 | 19.8M | 3.05B | 7ms | 143 | 79.5 |
| • **CoSNet-B**1 | ConvNet | 26 | 22.0M | 3.50B | 7ms | 167 | 79.9 |
| • **CoSNet-B**2 | ConvNet | 26 | 30.0M | 5.10B | 9ms | 111 | 81.3 |
| • **CoSNet-C**1 | ConvNet | 28 | 24.4M | 4.12B | 7ms | 143 | 80.0 |
| • **CoSNet-C**2 | ConvNet | 26 | 38.9M | 7.09B | 11ms | 90 | 82.1 |
| • **CoSNet-A**1**-PFF** | ConvNet | 38 | 12.7M | 1.93B | 7ms | 143 | 79.7 |
| • **CoSNet-B**0**-PFF** | ConvNet | 38 | 21.8M | 3.44B | 8ms | 125 | 80.6 |
| • **CoSNet-B**1**-PFF** | ConvNet | 38 | 25.6M | 4.08B | 8ms | 125 | 81.4 |
| • **CoSNet-B**2**-PFF** | ConvNet | 38 | 34.3M | 5.91B | 10ms | 100 | 82.7 |
| • **CoSNet-C**1**-PFF** | ConvNet | 42 | 27.3M | 4.75B | 8ms | 125 | 81.3 |
| • **CoSNet-C**2**-PFF** | ConvNet | 38 | 44.5M | 8.27B | 13ms | 77 | 83.7 |

lower latency. Even the CoSNet-A1-PFF variant is still relatively shallower than EfficientVit while delivering better accuracy.

**CoSNet *vs* DeiT (Touvron et al., 2021)** From Table 1, CoSNet-B1 is almost 50% less deep, has 23% fewer params, and runs 60% faster than DeiT Transformer while exhibiting slightly better accuracy. With PFF, CoSNet-B0-PFF performs better in terms of accuracy, depth, and runtime.

**CoSNet *vs* advanced mid-range ConvNets and Transformers** CoSNet-B2 is 72% less deeper, 55% faster, and 1.2% more accurate than the popular Swin Transformer (Liu et al., 2021). It is also 55%

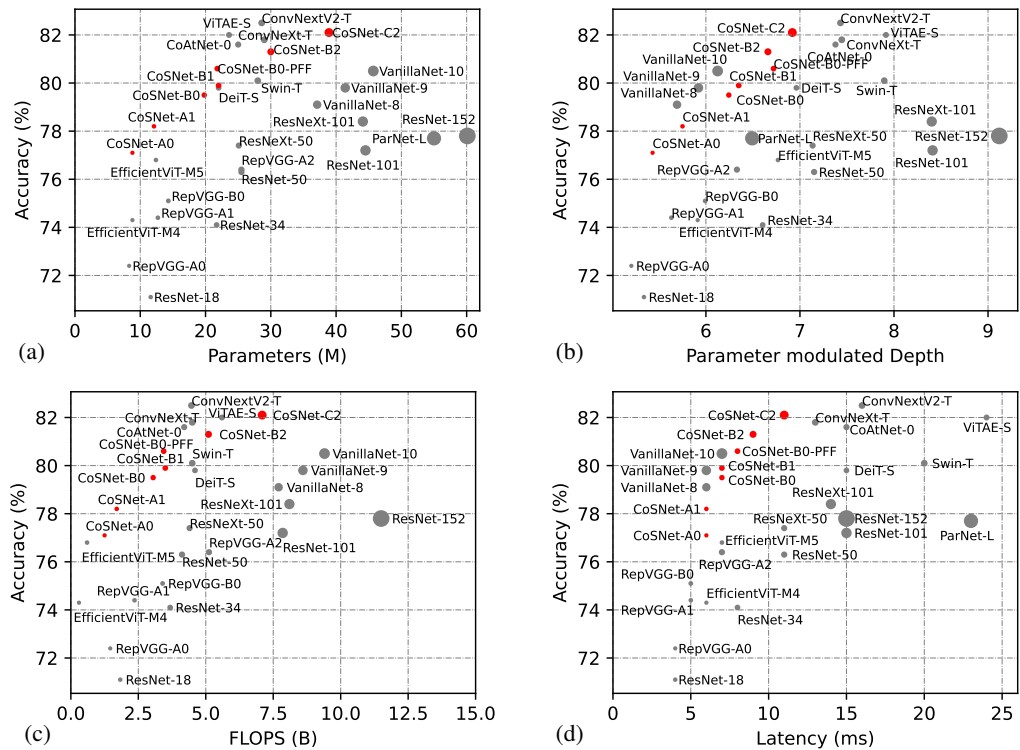

Figure 5: Comparing the proposed CoSNet with representative models. Models in '●' and '●' refers to CoSNet and existing models respectively. CoSNet has lower parameters, lower FLOPs, while depth of CoSNet is not unnecessarily large. The size of the circle is proportional to the parameter count.

less deeper, 30% faster with slightly lower accuracy than the popular ConvNeXt (Liu et al., 2022). Moreover, CoSNet-C2 rivals the latest ConvNext-v2-T (Woo et al., 2023) with similar accuracy but higher speed and smaller depth.

CoSNet-B2, C1, C2 models rivals advanced Transformers, such as ViTAE-S (Xu et al., 2021) and hybrid models, such as CoAtNet-0 (Dai et al., 2021b). With similar parameter counts and accuracy, our models show faster inference speed. The competitive tradeoffs offered by CoSNet show the significance of concise models.

## 4.2 COMPARISON WITH STANDARD CONVNETS

We show that CoSNet achieves efficiency in multiple aspects in a large spectrum of models while being simpler during training and inference and offering competitive trade-offs relative to the rival network. See Table 1 for the comparison. Figure 5 plots the trends regarding various aspects.

**CoSNet *vs* recent VanillaNet (Chen et al., 2023)**. CoSNet rivals recent ConvNet design, VanillaNet. VanillaNet is shallow and mainly focuses on latency. Our CoSNet-A0 shows similar latency at fewer parameters, fewer FLOPs, and high accuracy compared with VanillaNet-6 (Table 1).

**CoSNet *vs* recent ParNet (Goyal et al., 2021).** CoSNet outperforms recent non-deep ParNet that focuses on lower latency (Table 1 R4). CoSNet is uni-branched, while ParNet has multiple shallow branches which serialize the computations, thus making them deeper virtually.

**CoSNet *vs* RepVGG (Ding et al., 2021).** RepVGG offers a plain VGG-like (Simonyan & Zisserman, 2014) structure via Structural Reparameterization (SR). However, its training complexity is high due to a large number of parameters and three branches at each layer (Figure 1f). Hence, we show its performance with and without SR.

Compared with the RepVGG family, CoSNet offers considerably lower complexity during training and testing, thanks to its parallel columnar convolutions. In addition, CoSNet has fewer parameters

Table 2: Comparison with VannilaNet Chen et al. (2023) in training.

| Architecture | #Depth↓ | #Epochs↓ | #Params↓ | #FLOPs↓ | Top-1 (%)↑ | Train Time Per Epoch ↓ | Train Time 300 Epochs ↓ |
|---|---|---|---|---|---|---|---|
| VanillaNet-6 Chen et al. (2023) | 6 | 300 | 32.0M | 6.00B | 76.36 | 8 minutes | 40 hours |
| VanillaNet-8 Chen et al. (2023) | 8 | 300 | 37.1M | 7.70B | 79.13 | 11 minutes | 55 hours |
| • **CoSNet-B1** | 26 | 300 | **19.8**M | **3.05**B | **79.50** | **5 minutes** | **25 hours** |

Table 3: CoSNet with SE-like modules Hu et al. (2018).

| Approach | #Epochs | #Depth | #Params | #FLOPs | Top-1 (%) |
|---|---|---|---|---|---|
| ResNet-50 + SE Hu et al. (2018) | 120 | 50 | 28.0M | 4.13B | 76.85 |
| ResNet-50 + CBAM Woo et al. (2018) | 120 | 50 | 28.0M | 4.13B | 77.34 |
| • **CoSNet-B1** | 120 | **26** | **19.2**M | **3.05**B | **76.77** |
| • **CoSNet-B1** + SE Hu et al. (2018) | 120 | **26** | **20.1**M | **3.10**B | **77.85** |
| ResNet-50 + AFF Dai et al. (2021a) | 160 | 50 | 30.3M | 4.30B | 79.10 |
| ResNet-50 + SKNet Li et al. (2019) | 160 | 50 | 27.7M | 4.47B | 79.21 |
| • **CoSNet-C1** + SE Hu et al. (2018) | 160 | **28** | **25.0**M | **4.13**B | **79.51** |

and fewer FLOPs while offering similar speeds with higher accuracy. For instance, CoSNet-B2 is better than RepVGG-B3 at similar depth, 73% fewer parameters, 80% lesser FLOPs while running faster. This shows the significance of parallel columns of CoSNet that during model scaling, parameter count does not grow rapidly.

**CoSNet *vs* EfficientNet (Tan & Le, 2019).** Although we do not aim for a mobile regime in this paper, we show that having fewer parameters and FLOPs does not guarantee faster speeds. As shown in Table 1, EfficientNet-B0 has 50% fewer parameters and 77% fewer FLOPs, but is 50% deeper, and runs 37% slower. By exploring the design space, CoSNet can be extended to the mobile regime.

**CoSNet *vs* ResNet (He et al., 2016) family.** As shown in Table 1, CoSNet-A0 is 6% more accurate, has 25% fewer parameters, shows similar runtime, and shows 31% fewer FLOPs than ResNet-18 although CoSNet has 6 more layers. Similarly, in contrast to ResNet-34, it is more accurate by 3% with 59% fewer parameters, 66% fewer FLOPs, and 23% less layers, while it is fast by 37%. ResNet-50 is the widely employed backbone in downstream tasks (Carion et al., 2020; Goyal et al., 2017; He et al., 2017; Ren et al., 2015) due to its affordability regarding representation power, FLOPs, depth, and accuracy. Table 1 shows that CoSNet-B0 surpasses ResNet-50 while being 50% shallower, 22% fewer parameters, 25% fewer FLOPs, and 40% faster.

**CoSNet *vs* bigger ResNet (He et al., 2016) and ResNeXt (Xie et al., 2017) models.** As shown in Table 1, CoSNet-C1 is better than bigger variants of ResNet, which serves as backbones for cutting-edge works (Carion et al., 2020; Li et al., 2022). Our CoSNet outperforms them in various aspects while being 72% and 82% less deep relative to ResNet-101 and ResNet-152, respectively. CoSNet also runs faster by 50% in 50% fewer parameters and FLOPs. In addition, despite being smaller than ResNeXt (Xie et al., 2017), CoSNet-C1 outperforms it in various aspects. Overall CoSNet-C1 is 50% less deeper than ResNeXt-50 while running 50% faster at 6% fewer FLOPs, 2% fewer parameters while being more accurate. In contrast to ResNeXt-101, CoSNet-C2 is 75% less deeper, 11% fewer parameters, 12% fewer FLOPs, and 35% faster at a higher accuracy.

### 4.3 ADDITIONAL EXPERIMENTS

**CoSNet has small training walltime.** We provide an additional comparison with the recent ConvNet design, VanillaNet (Chen et al., 2023), under training settings. Table 2 shows that despite VanillaNet being a shallow network, it has a high training time. We speculate that the large number of channels in the deeper layers of VanillaNet slows down batch processing at large batch sizes. In CoSNet, parallel columnar convolutions and controlled parameter growth in the deeper layers counter this issue, leading to lower training time.

**CoSNet is seamlessly compatible with SE-like (Hu et al., 2018) modules.** Table 3 shows the results when CoSNet is used in conjunction with Squeeze and Excitation (SE) like modules (Hu et al., 2018). It outperforms recent attention mechanism (AFF (Dai et al., 2021a), SKNet (Li et al., 2019), and CBAM (Woo et al., 2018)) applied to ResNet-50.

Table 4: CoSNet in state-of-the-art Detection Transformers (DETR) Li et al. (2022) @12 epochs setting.

| Method | #Params | #FPS | AP | $AP_{50}$ | $AP_{75}$ | $AP_S$ | $AP_M$ | $AP_L$ |
|---|---|---|---|---|---|---|---|---|
| DN-DETR-ResNet50 Li et al. (2022) | 44M | 24 | 38.3 | 59.1 | 41.0 | 17.3 | 42.4 | 57.7 |
| • **DN-DETR-CoSNet-C2** | 56M | **25** | 39.2 | 60.0 | 41.9 | 18.1 | 43.0 | 59.1 |

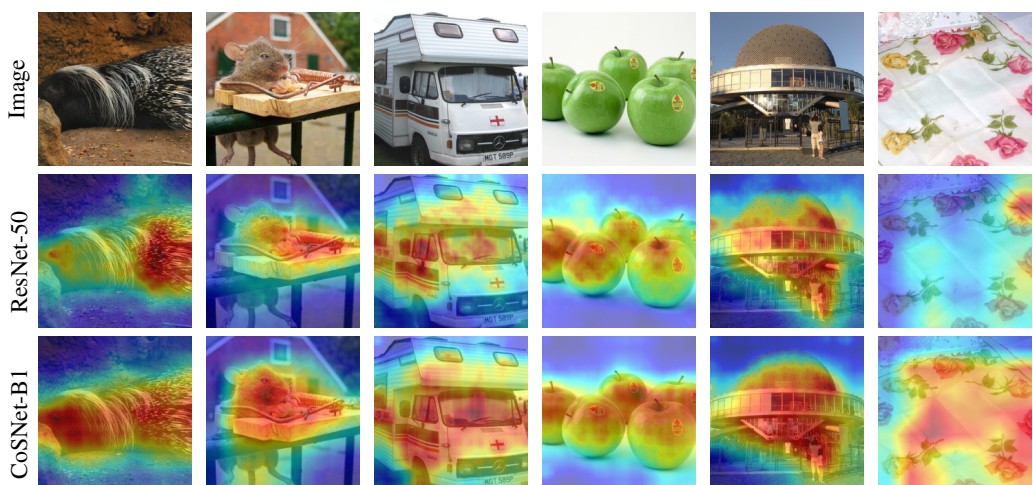

Figure 6: CAM Srinivas & Fleuret (2019) visualizations. Notably, CoSNet attends the class regions more accurately than the baseline.

## 4.4 CoSNet in State-of-the-art Detection Transformer

We apply CoSNet to state-of-the-art object Detection Transformer, DN-DETR (Li et al., 2022) to demonstrate the effectiveness of CoSNet in the downstream task. We experiment on MS-COCO (Lin et al., 2014) benchmark and utilized DN-DETR's default training settings.

Table 4 shows that DN-DETR with CoSNet improves the inference speed and average precision compared to the DN-DETR with ResNet-50 backbone. By further optimizing the DETR hyperparameters, CoSNet can be configured to deliver better performance.

## 4.5 Visualization of Attention

To comprehend CoSNet's better performance, we investigate its class activation maps (CAM) on ImageNet (Deng et al., 2009) validation set. We use CAM output from popular Full-Grad-CAM (Srinivas & Fleuret, 2019) for a given class. CAM visualizations of ResNet-50 and CoSNet-B1 are shown in Figure 6. It can be seen that CoSNet, despite being 50% shallower than ResNet, is better at learning to attend regions of the target class relative to the baseline.

## 5 Conclusion

We propose *CoSNet*, which revisits ConvNet design based on multiple aspects for concise models. CoSNet is based on our parallel columnar convolutions and input replication concepts to be efficient in parameters, FLOPs, accuracy, latency, and training duration. Through extensive experimentation and ablations, we show that CoSNet rivals many representative ConvNets and ViTs such as ResNet, ResNeXt, RegNet, RepVGG, and ParNet, VanillaNet, DeiT, EfficientViT while being shallower, faster, and being architecturally simpler.

**Future work.** CoSNet is open for improvement. In this paper, we have built a simple template architecture that can further evolve like ConvNext (Liu et al., 2022). For instance, a comprehensive design space of CoSNet including mobile regime can be explored, similar to RegNet (Radosavovic et al., 2020). Besides, layer merging post-training, shown in VanillarNet (Chen et al., 2023), can be utilized to develop shallower variants of CoSNet. In addition to that, CoSNet can also be married with a Transformer attention mechanism like (Dai et al., 2021b) or (Liu et al., 2023).

**Acknowledgements.** Jaesik Park was supported by MSIT grant (RS-2021-II211343: AI Graduate School Program at Seoul National University (5%) and 2023R1A1C200781211 (95%))

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

APPENDIX

## A  COSNET INSTANCES

Table A1 shows CoSNet instances configurations mentioned in the main paper.

Table A1: CoSNet instances Configurations.

| Model | $P_c$ | | | | $N$ | | | | $l$ | | | | $M$ | | | | #Depth | #Params | #FLOPs |
|---|---|---|---|---|---|---|---|---|---|---|---|---|---|---|---|---|---|---|---|
| • CoSNet-A0 | 256 | 512 | 1024 | 2048 | 16 | 32 | 64 | 128 | 3 | 4 | 6 | 3 | 1 | 1 | 1 | 1 | 26 | 8.8M | 1.25B |
| • CoSNet-A1 | 256 | 512 | 1024 | 2048 | 16 | 32 | 64 | 128 | 3 | 4 | 6 | 3 | 4 | 4 | 4 | 4 | 26 | 12.1M | 1.77B |
| • CoSNet-B0 | 256 | 512 | 1024 | 2048 | 32 | 64 | 128 | 256 | 3 | 4 | 6 | 3 | 4 | 4 | 4 | 4 | 26 | 19.8M | 3.05B |
| • CoSNet-B1 | 256 | 512 | 1024 | 2048 | 32 | 64 | 128 | 256 | 3 | 4 | 6 | 3 | 5 | 5 | 5 | 5 | 26 | 22.6M | 3.51B |
| • CoSNet-B2 | 256 | 512 | 1024 | 2048 | 32 | 64 | 128 | 256 | 3 | 4 | 6 | 3 | 4 | 4 | 16 | 4 | 26 | 30.0M | 5.1B |
| • CoSNet-C1 | 256 | 512 | 1024 | 2048 | 48 | 80 | 144 | 272 | 4 | 4 | 6 | 4 | 4 | 4 | 4 | 4 | 28 | 24.4M | 4.12B |
| • CoSNet-C2 | 256 | 512 | 1024 | 2048 | 48 | 80 | 144 | 272 | 3 | 4 | 6 | 3 | 6 | 6 | 16 | 6 | 26 | 38.9M | 7.09B |

## B  ABLATION STUDY

**Varying M and N.** Table A2 demonstrates the effect of varying $N$ and $M$ (R0-R5). We first fix the values of $N$ and vary $M$ (R0-R5), and then vary $M$ while fixing $N$ (R0 $\leftrightarrow$ R3, R1 $\leftrightarrow$ R4, R2 $\leftrightarrow$ R5). For fixed $N$, accuracy improves by increasing $M$, and the same effect is seen by fixing $M$ while varying $N$. It can be noticed that parameters, FLOPs can be controlled by changing the $M$ (R1 $\leftrightarrow$ R2, R4 $\leftrightarrow$ R5), which directly reflects accuracy.

**Effect of `PCC`.** We compare instances having different $N, M$, but have similar parameters and FLOPs budget, for instance, R1 $\leftrightarrow$ R2, R1 $\leftrightarrow$ R3, Table A2. Noticeably, $R2$ with 5 `PCC` is better by 0.36% in accuracy, only at 1.1M more parameters relative to R1. Similarly, $R$1 is better by 0.28% in accuracy, only at 0.8M more parameters relative to R3. It shows that multiple `PCC`s facilitates improved accuracy in just a fraction of parameters and FLOPs. Moreover, if comparing R9 (a deeper model) with R2, R2 achieves 0.13% more accuracy in 0.2M fewer parameters and 0.17B fewer FLOPs. It shows the advantage of *having multiple convolutional modules while being shallower*.

**Varying l.** The impact of varying $l$ is shown in R9, Table A2. It can be seen that *going deeper is not necessary* because a shallower version with same parameters (R2) is more accurate. Moreover, increased depth causes increased latency in R9. Therefore, we stick to $20 - 40$ layers of depth.

**Group Convolution or ResNext-like Xie et al. (2017) Setting** We also conduct additional experiments where each `PCC` is followed by a $1 \times 1$ convolution as done in group convolutions while keeping depth and parameters constant. We observe a 1% accuracy drop. This indicates that frequent fusion similar to ResNeXt is not necessary.

**Effect of Shallow Projections in `PCC`.** R6-R9, Table A2 shows this analysis. For the shallower model, the residual connection shows only minor improvement (0.09%), however, for the deeper model, the effect of residual connections is noticeable (0.70%).

**Effect of Deep Projections (`DP`).** We train an CoSNet instance in three ways: *First*, remove `DP` entirely, *Second*, use `DP` without `pooling`, and *Third*, `DP` with `pooling`. See Table A2 for the analysis. It can be noticed that without `DP` (R10), the model suffers with heavy accuracy loss of $\sim 0.54\%$ relative to when `DP` is used without `pooling` (R11). Moreover, when using `DP` with `pooling` (R12), accuracy improves, i.e., 1.22% and 1.76% relative to R11 and R10, respectively, because `pooling` provides more spatial context to the $1 \times 1$ $L_p$ layer by summarizing the neighborhood.

**Effect of using very small $N$ to compare with Group Convs Zhang et al. (2018) and depthwise Conv-like Howard et al. (2017) structure.** From Table A2, R13-14, it can be seen that when in deeper layers, $N$ is restricted to a very smaller value while keeping parameters or FLOPs the same, accuracy decreases considerably. This is because of the reason mentioned in the main paper (Sec."Fuse-Once") that too few connections restrict the learning ability of neurons. Hence, they need frequent fusion similar to GroupWise and Depthwise convolution methods, but it increases depth. To avoid that, we increase $N$ as we go deep down in CoSNet, which does not require frequent fusion due

Table A2: Effect of parallel columnar convolution (PCC), # of kernels $N$, # of layers $l$, # of parallel convolutions $M$, and Deeper projections (DP). Values of $M,N,l$ are for each of the four CoSNet stages. Ablations are conducted at 120 epochs.

| Row | N | | | | l | | | | M | | | | #Depth | DP | Residual in PCC | #Params | #FLOPs | Top-1 (%) |
|---|---|---|---|---|---|---|---|---|---|---|---|---|---|---|---|---|---|---|
| R0 | ●16 | 32 | 64 | 128 | 3 | 4 | 6 | 3 | 1 | 1 | 1 | 1 | 26 | ✓ | ✓ | 8.80M | 1.25B | 74.45 |
| R1 | ●16 | 32 | 64 | 128 | 3 | 4 | 6 | 3 | 4 | 4 | 4 | 4 | 26 | ✓ | ✓ | 12.1M | 1.77B | 75.65 |
| R2 | ●16 | 32 | 64 | 128 | 3 | 4 | 6 | 3 | 5 | 5 | 5 | 5 | 26 | ✓ | ✓ | 13.2M | 1.95B | 76.01 |
| R3 | ●32 | 64 | 128 | 256 | 3 | 4 | 6 | 3 | 1 | 1 | 1 | 1 | 26 | ✓ | ✓ | 11.3M | 1.65B | 75.37 |
| R4 | ●32 | 64 | 128 | 256 | 3 | 4 | 6 | 3 | 4 | 4 | 4 | 4 | 26 | ✓ | ✓ | 19.8M | 3.05B | 76.76 |
| R5 | ●32 | 64 | 128 | 256 | 3 | 4 | 6 | 3 | 5 | 5 | 5 | 5 | 26 | ✓ | ✓ | 22.6M | 3.51B | 77.01 |
| R6 | ●32 | 64 | 128 | 256 | 3 | 4 | 6 | 3 | 1 | 1 | 1 | 1 | 26 | ✓ | ✗ | 11.3M | 1.65B | 75.28 |
| R7 | ●32 | 64 | 128 | 256 | 3 | 4 | 6 | 3 | 1 | 1 | 1 | 1 | 26 | ✓ | ✓ | 11.3M | 1.65B | 75.37 |
| R8 | ●32 | 64 | 128 | 256 | 4 | 5 | 20 | 3 | 1 | 1 | 1 | 1 | 44 | ✓ | ✗ | 13.4M | 2.12B | 75.18 |
| R9 | ●32 | 64 | 128 | 256 | 4 | 5 | 20 | 3 | 1 | 1 | 1 | 1 | 44 | ✓ | ✓ | 13.4M | 2.12B | 75.88 |
| R10 | ●32 | 64 | 128 | 256 | 3 | 4 | 6 | 3 | 1 | 1 | 1 | 1 | 26 | ✗ | ✓ | 8.5M | 1.29B | 73.61 |
| R11 | ●32 | 64 | 128 | 256 | 3 | 4 | 6 | 3 | 1 | 1 | 1 | 1 | 26 | w/o. Pooling | ✓ | 9.8M | 1.44B | 74.15 |
| R12 | ●32 | 64 | 128 | 256 | 3 | 4 | 6 | 3 | 1 | 1 | 1 | 1 | 26 | w. Pooling | ✓ | 9.8M | 1.44B | 75.37 |
| R13 | ●32 | 64 | 128 | 256 | 3 | 4 | 6 | 3 | 4 | 4 | 4 | 4 | 26 | ✓ | ✓ | 19.8M | 3.51B | 76.76 |
| R14 | ●32 | 32 | 32 | 32 | 3 | 4 | 6 | 3 | 4 | 8 | 16 | 32 | 26 | ✓ | ✓ | 18.4M | 3.42B | 71.20 |

Table A3: Effect of batch size on the baselines and CoSNet in the context.

| Architecture | Type | Batch Size | #Depth ↓ | #Params ↓ | FLOPs ↓ | Latency ↓ | FPS ↑ | Top-1 (%) ↑ |
|---|---|---|---|---|---|---|---|---|
| EfficientNet-B0 Tan & Le (2019) | ConvNet | 256 | 49 | 5.3M | 0.40B | 8ms | 125 | 75.1 |
| EfficientNet-B0 Tan & Le (2019) | ConvNet | 2048 | 49 | 5.3M | 0.40B | 8ms | 125 | 77.1 |
| EfficientViT-M5 Liu et al. (2023) | Transformer | 256 | 70 | 12.4M | 0.60B | 7ms | 142 | 76.8 |
| EfficientViT-M5 Liu et al. (2023) | Transformer | 2048 | 70 | 12.4M | 0.60B | 7ms | 142 | 77.1 |
| ●**CoSNet-A**0 | ConvNet | 256 | 26 | 8.8M | 1.25B | 6ms | 167 | 77.1 |
| ●**CoSNet-A**1**-PFF** | ConvNet | 256 | 38 | 12.7M | 1.93B | 7ms | 143 | 79.7 |
| ConvNeXt-T Liu et al. (2022) | ConvNet | 256 | 59 | 29.0M | 4.50B | 13ms | 77 | 81.8 |
| ConvNeXt-T Liu et al. (2022) | ConvNet | 4096 | 59 | 29.0M | 4.50B | 13ms | 77 | 82.1 |
| ●**CoSNet-C**2 | ConvNet | 256 | 26 | 38.9M | 7.09B | 11ms | 90 | 82.1 |
| ●**CoSNet-B**2**-PFF** | ConvNet | 256 | 38 | 34.3M | 5.91B | 10ms | 100 | 82.7 |

to a sufficiently large number of neuron connections. Thus, we fuse only once, eliminating the need for fusion $1 \times 1$ layers, thus smaller depth and lower latency.

## C  THE EFFECT OF BATCH SIZES OF THE BASELINE APPROACHES.

In the literature, some baselines are trained with larger batch sizes (above 1024), but others have been trained at a much smaller batch size (256). Therefore, we retrained high batch size baselines with 256 batch sizes to avoid getting biased conclusions about the effects of large batch sizes. Such results with 256 batch size are carefully reported in Table 1.

In this section, we present the results of the baselines with larger batch sizes in Table A3. As widely studied, the baseline approaches Tan & Le (2019); Liu et al. (2023; 2022) show improved accuracy. Interestingly, it can be noticed that CoSNet trained with a 256 batch size can compete with state-of-the-art approaches trained with a larger batch size. This shows the utility of obtaining higher accuracies in resource-constrained training scenarios (i.e., limited memory to fit 4096 batch, etc.).

## D  ADDITIONAL RESULTS

Table A4 shows results on RetinaNet $x$1 Lin et al. (2017) detection pipeline. It can be seen that, for a comparable vision transformer backbone, CoSNet performs better. We also provide semantic segmentation results for the popular PSPNet pspnet semantic segmentation framework. It can be seen that CoSNet performs better than the baselines.

Table A4: CoSNet in RetinaNet *x*1 Lin et al. (2017) object detection framework.

| Method | #Depth | #Params | AP | $AP_S$ | $AP_M$ | $AP_L$ |
|---|---|---|---|---|---|---|
| EfficientViT-M4 Liu et al. (2023) | 42 | 8.8M | 32.7 | 17.6 | 35.3 | 46.0 |
| ● **CoSNet-A0** | **26** | 8.8M | 34.3 | 19.1 | 38.0 | 49.1 |

Table A5: CoSNet in PSPNet Zhao et al. (2017) semantic segmentation framework.

| Method | #Params | mIoU | FPS |
|---|---|---|---|
| RepVGG-B1g2 Ding et al. (2021) | 41.36M | 78.88 | 13 |
| ResNet-50 | 25.5M | 77.17 | 13 |
| ● **CosNet-B1** | 22.0M | 79.05 | 17 |

## E  TRAINING SETTING

We train models in PyTorch Paszke et al. (2019) using eight NVIDIA A40 GPUs.

## F  PYTORCH CODE

All codes shall be open-sourced in PyTorch Paszke et al. (2019) post the review process. Here, we provide a code snippet of a CoSNet-Unit. Please see until the end of this document.

```python
class InputReplicator(nn.Module):
    def __init__(self, M):
        super(InputReplicator, self).__init__()

        # number of Parallel Columnar Convolutions
        self.M = M

    def forward(self, ip):
        x = ip.repeat(1, self.M, 1, 1)
        return x

class CoSNetUnit(nn.Module):
    def __init__(self, n_ip):
        super(CoSNetUnit, self).__init__()

        self.n_op_Lf = 256 # 512, 1024, 2048
        self.N = 32
        self.stride = 2
        self.l = 3 # 4, 6, 3]
        self.M = 4# 4, 4, 4]

        n_op_Ls = int(self.n_op_Lf / 4)

        self.conv_ls = nn.Conv2d(n_ip, n_op_Ls, 1, 1, 0, bias=False)
        self.bn_ls = nn.BatchNorm2d(n_op_Ls)
        self.act_ls = nn.SiLU(True)

        self.IR = InputReplicator(self.M)

        # we limit the n_op of last PCC layer so that the parameters of the 1x1 expansion layer
        # do not grow overly large if number of columns is very big
        # as a rule of thumb, we set it nearly equal to  n_op / 4
        self.n_op_pcc_last = int(round(n_op_Ls / self.M)) * self.M

        self.conv_pcc = nn.ModuleList()
        self.bn_pcc = nn.ModuleList()
        self.act_pcc = nn.ModuleList()

        self.conv_pcc.append(nn.Conv2d(n_op_Ls * self.M, self.N * self.M, 3, self.stride, 1,
                                    groups=self.M, bias=False))
        self.bn_pcc.append(nn.BatchNorm2d(self.N * self.M))
        self.act_pcc.append(nn.SiLU(True))

        for i in range(self.l-2):
            self.conv_pcc.append(nn.Conv2d(self.N * self.M, self.N * self.M, 3, 1, 1,
                                        groups=self.M, bias=False))
            self.bn_pcc.append(nn.BatchNorm2d(self.N * self.M))
            self.act_pcc.append(nn.SiLU(True))
```

```python
52          self.conv_pcc.append(nn.Conv2d(self.N * self.M, self.self.n_op_pcc_last, 3, 1, 1,
53                                        groups=self.M, bias=False))
54          self.bn_pcc.append(nn.BatchNorm2d(self.n_op_pcc_last))
55          self.act_pcc.append(nn.SiLU(True))
56
57          self.conv_lf = nn.Conv2d(self.n_op_pcc_last, self.n_op_Lf, 1, 1, 0, bias=False)
58          self.bn_lf = nn.BatchNorm2d(self.n_op_Lf)
59          self.act_lf = nn.SiLU(True)
60
61
62          self.conv_lp = nn.Conv2d(n_ip, self.n_op_Lf, 1, 2, 0, bias=False)
63          self.bn_lp   = nn.BatchNorm2d(self.n_op_Lf)
64
65      def forward(self, ip):
66          x = self.act_ls(self.bn_ls(self.conv_ls(ip)))
67          x = self.IR(x)
68
69          x = self.act_pcc[0](self.bn_pcc[0](self.conv_pcc[0](x)))
70
71          for i in range(1, self.l - 2):
72              y = self.bn_pcc[i](self.conv_pcc[i](x))
73              x = self.act_pcc[i](x + y)
74
75          # Last pccN needs to handled with care because n_op for last pcc may not match
76          # with n_op of the previous pcc layer
77          # and thus an identity residual connection is not possible
78          # In other words, a residual connection will be used iff n_op of all pcc layers
79          # is same
80          if (self.N * self.M == self.n_op_pcc_last):
81              idx = self.l - 1
82              y = self.bn_pcc[idx](self.conv_pcc[idx](x))
83              x = self.act_pcc[idx](x + y)
84          else:
85              idx = self.l - 1
86              x = self.act_pcc[idx](self.bn_pcc[idx](self.conv_pcc[idx](x)))
87
88          x = self.bn_lf(self.conv_lf(x))
89
90          z = F.avg_pool2d(ip, 3, 2, 1)
91          z = self.bn_lp(self.conv_lp(z))
92
93          return self.act_lf(x + z)
```

## G  COMPLETE NETWORK VISUALIZATION

We also visualize the complete architecture of CoSNet-B1 variant and have put it in the context of ResNet-like models. We have plotted ResNet-50 variant. Please see Figure A1.

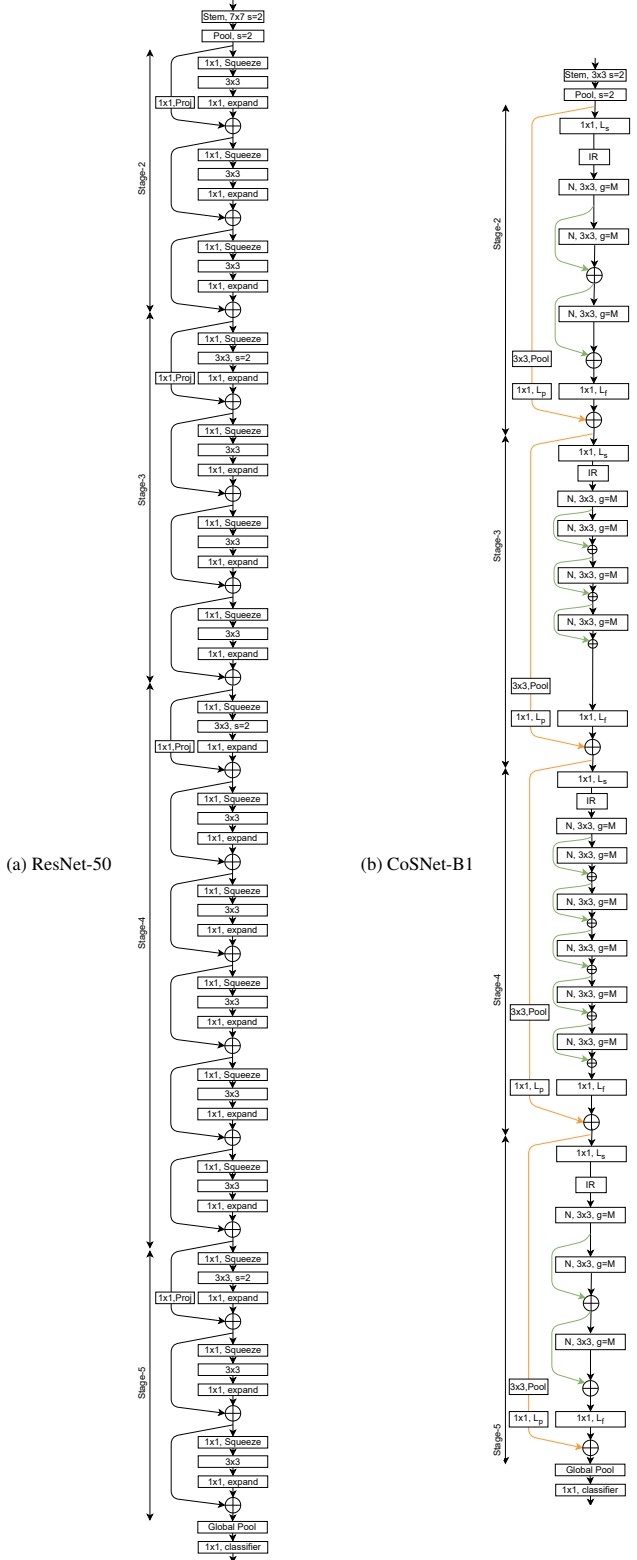

Figure A1: Illustration of (a) ResNet-50 He et al. (2016) network, and (b) CoSNet-B1. It must be noted that by merely replacing the residual bottleneck-based stages of ResNet with the proposed CoSNet-unit, our CoSNet variant becomes roughly 50% less deep, has 22% fewer parameters, 25% fewer FLOPs, and runs 40% faster. It shows the utility of CoSNet design from an efficiency perspective in multiple aspects.

