# OpenReview forum: "Designing Concise ConvNets with Columnar Stages"
_ICLR.cc/2025/Conference — ICLR 2025 Poster_

### Official Review · Reviewer_LJPJ · 2024-11-01

**Soundness:** 3
**Presentation:** 3
**Contribution:** 3
**Rating:** 6
**Confidence:** 5

**Summary:**

This paper introduces a refreshing ConvNet macro design called Columnar Stage Network (CoSNet) with smaller depth, low parameter count, low FLOPs, and attention-less operations.
Its comprehensive evaluations show that CoSNet rivals many renowned ConvNets and Transformer designs under resource-constrained scenarios.

**Strengths:**

1. The motivation is reasonable.
2. The result is comparable with the state-of-the-art.
3. The paper is easy to understand.

**Weaknesses:**

1. Typo: VanillaNet is published in NeurIPS 2023, instead of 2024.
2. Lack of some new comparison methods, all models were published in 2023 and even earlier.
   The author should provide more comparisons like InceptionNeXt[1] and UniRepLKNet [2].
3. The Top-1 accuracy of EfficientNet-B0 is 76.3 [3] or 77.1 [4], but the author gives a much poorer result of 75.1.
   Similar problems also happen on ConvNeXt-T (82.1 in [5] but 81.8 in this paper) and EfficientViT-M5 (77.1 in [6] but 76.8 in this paper, and 522M FLOPs in [6] and 600M in this paper).


[1] InceptionNeXt: When Inception Meets ConvNeXt. CVPR 2024
[2] UniRepLKNet: A Universal Perception Large-Kernel ConvNet for Audio, Video, Point Cloud, Time-Series and Image Recognition. CVPR 2024
[3] EfficientNet: Rethinking Model Scaling for Convolutional Neural Networks. ICML 2019
[4] EfficientNet: Rethinking Model Scaling for Convolutional Neural Networks. arXiv 2019
[5] A ConvNet for the 2020s. CVPR 2022
[6] EfficientViT: Memory Efficient Vision Transformer with Cascaded Group Attention. CVPR 2023

**Questions:**

As listed in Weaknesses

---

### Official Review · Reviewer_pdHY · 2024-11-02

**Soundness:** 3
**Presentation:** 3
**Contribution:** 3
**Rating:** 6
**Confidence:** 3

**Summary:**

This paper proposes a simple and concise structure called CoSNet, which has smaller depth, low parameter count, low FLOPs, and attention less operations, well suited for resource-constrained deploy. The work presents a range of experiments that sufficiently support its claims. It is very  interesting for readers.

Overall, it is a good read. The manuscript might get better if a few suggestions (given below) are incorporated.

**Strengths:**

1. The writing is easy to read and clearly explains everything in the paper.
2. The experimental result is good compared to the previous works. Empirically, the method seems to offer strong accuracy, compared to existing methods with similar architectures.

**Weaknesses:**

1. Some details are missing.  For example, how is the value of parallel convolution M determined? I think that different values of M will affect the performance. Please explain this details in the text. Other minor issues, such as Section 3.4 is missing in Figure 2 (c), and you should add it.
2. How is the design like "input replication" to improving performance for example? Authors need to give some details in the manuscript.
3. The related work is comprehensive. However, the authors only highlight the salient features of the previous works that they apply in their network. The manuscript can benefit from discussing shortcomings of the existing methods as research gaps in the section "Related Work".

**Questions:**

1. Some details are missing.  For example, how is the value of parallel convolution M determined? I think that different values of M will affect the performance. Please explain this details in the text. Other minor issues, such as Section 3.4 is missing in Figure 2 (c), and you should add it.
2. How is the design like "input replication" to improving performance for example? Authors need to give some details in the manuscript.
3. The related work is comprehensive. However, the authors only highlight the salient features of the previous works that they apply in their network. The manuscript can benefit from discussing shortcomings of the existing methods as research gaps in the section "Related Work".

---

### Official Review · Reviewer_nNEm · 2024-11-05

**Soundness:** 3
**Presentation:** 3
**Contribution:** 3
**Rating:** 6
**Confidence:** 4

**Summary:**

This paper introduces Columnar Stage Network (CoSNet) to deploy parallel conv units with fewer kernels, and reduce the 1x1 conv layers. To optimize the model efficiency, this paper follows the design objectives to reduce depth and branching, as well as to control the parameter growth, maintain computation intensity and uniform primitive operations.

**Strengths:**

This paper revisits some of the fundamental design ideas in conv net and proposed some interesting ideas.
1. Shallow-deep projection is quite interesting. This inherits ideas from ResNet and expands to deep connection.
2. It achieves competitive performances (accuracy and latency) with reduced network depth and parameters counts.
3. It also introduces a pairwise frequent fusion (PFF) to fuse information across different columns.

**Weaknesses:**

Please refer to the questions section, where some clarity or more experiments would be great.

**Questions:**

1. what's the difference between group conv and parallel columnar conv?
2. How much of efficiency and accuracy gain can be translated downstream to segmentation and detection work?
3. What's the memory overhead with the input replication?
4. How much depth is required for deep projection to have significant impact?
5. I see some similarity between PFF and self-attention modules in transformer. What's the performance like if using some fusing all columns simultaneously?

---

### Meta-Review · Area_Chair_hp5i · 2024-12-17

**Metareview:**

This paper presents CoSNet, a series of network architectures of convolutional architectures to improve the efficiency of deep neural networks. At the core of this work lies a lot of model designs and adjustments; finally, an efficient architecture is obtained and works well in image classification.

The strengths of this work include:
* The proposed architecture seems simple and effective.
* Latency is tested which means the network enjoys runtime efficiency.

The weaknesses of this work include:
* The architecture has not been thoroughly tested for downstream tasks such as detection and segmentation.
* It seems difficult for the designed architecture to adapt to the transformer architecture for multimodal understanding; more importantly, given the prosperity of the vision transformer ecosystem, the advantage of model design will be largely reduced because vision transformers can benefit from a lot of other algorithms and pre-training data and/or pre-trained models.

All reviewers suggest borderline acceptance. Although the paper has clear limitations (see above), the AC finds no fatal reasons to overturn the reviewers' recommendation. However, the AC will not be upset if the SAC eventually decides to reject it. **If this paper is accepted, the AC strongly recommends the authors to add results on downstream recognition tasks in the final version.**

**Additional Comments On Reviewer Discussion:**

The paper receives an initial rating of 5/5/6. After the rebuttal and discussion, all reviewers responded and two negative reviewers raised the scores from 5 to 6. The AC finds no fatal reasons to reject the paper.

---

### Decision · Program_Chairs · 2025-01-22

Accept (Poster)